# The Offline-Frontier Shift: Diagnosing Distributional Limits in Generative Multi-Objective Optimization

**Stephanie Holly**[*]
LIT AI Lab and Institute for Machine Learning
JKU Linz, Austria

**Alexandru-Ciprian Zăvoianu**
School of Computing, Engineering and Technology
RGU, Aberdeen, Scotland

**Siegfried Silber**
Linz Center of Mechatronics GmbH
Linz, Austria

**Sepp Hochreiter**
LIT AI Lab and Institute for Machine Learning
JKU Linz, Austria

**Werner Zellinger**
LIT AI Lab and Institute for Machine Learning
JKU Linz, Austria

## Abstract

Offline multi-objective optimization (MOO) aims to recover Pareto-optimal designs given a finite, static dataset. Recent generative approaches, including diffusion models, show strong performance under hypervolume, yet their behavior under other established MOO metrics is less understood. We show that generative methods systematically underperform evolutionary alternatives with respect to other metrics, such as generational distance. We relate this failure mode to the offline-frontier shift, i.e., the displacement of the offline dataset from the Pareto front, which acts as a fundamental limitation in offline MOO. We argue that overcoming this limitation requires out-of-distribution sampling in objective space (via an integral probability metric) and empirically observe that generative methods remain conservatively close to the offline objective distribution. Our results position offline MOO as a distribution-shift–limited problem and provide a diagnostic lens for understanding when and why generative optimization methods fail.

## 1 Introduction

Multi-objective optimization (MOO) aims to recover optimal trade-offs among competing objectives and has been extensively studied in evolutionary computation (Zitzler & Thiele, 1999; Deb et al., 2002; Ishibuchi et al., 2015). In many real-world settings, objective evaluations are expensive or unavailable, and optimization must be performed using only an offline dataset (Trabucco et al., 2022; Xue et al., 2024; Chasparis et al., 2016). Recent advances in generative modeling motivate their use in offline MOO, enabling efficient sampling from high-dimensional spaces to produce diverse, high-quality designs.

Despite their potential, the behavior and limitations of generative models in MOO are not well understood. Prior evaluations have focused primarily on hypervolume (Xue et al., 2024; Yuan et al., 2025), offering limited insight into when and why generative models fail under other metrics. In this work, we extend the evaluation of generative methods across additional metrics, with a focus on the role of the offline dataset and out-of-distribution designs. Our evaluations witness the discrepancy between the fact that generative models have been developed to reproduce the data distribution, while offline optimization requires out-of-distribution exploration in objective space.

---

[*]Corresponding author holly@ml.jku.at.

Our contributions are:

- We show that, despite strong hypervolume performance, generative methods (Yuan et al., 2025; Annadani et al., 2025) underperform evolutionary methods on multiple benchmarks (Xue et al., 2024) in terms of alternative metrics, including generational distance and inverted generational distance (Section 5.1).

- We introduce the *offline-frontier shift* as a fundamental limiting factor (Lemma 1) and show that increasing shifts correspond to a stronger performance degradation for generative methods than for evolutionary alternatives (Figure 1 and Section 5.2).

- Our empirical evaluations show that current generative methods tend to remain closer to the offline distribution compared to evolutionary methods in terms of a probability distance (Section 5.1).

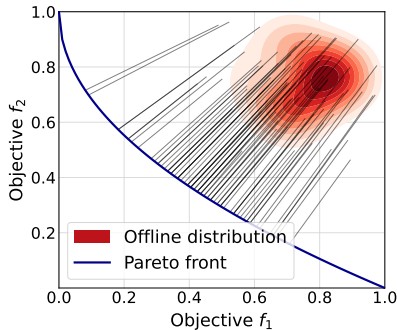 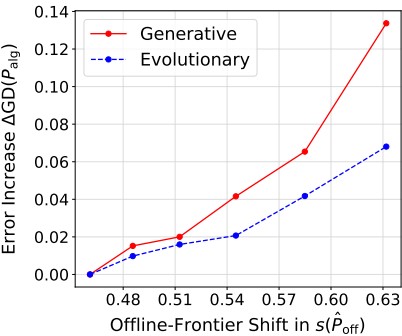

Figure 1: Left: offline-frontier shift (average projection distance; projections in gray). Right: error increase, showing that larger shifts lead to stronger degradation, especially for generative methods.

## 2 PROBLEM

**Offline MOO:** Consider a design space $\mathcal{X} \subset \mathbb{R}^d$ and a sequence $f_1, \ldots, f_m : \mathcal{X} \to \mathbb{R}$ of objective functions. Given access only to an offline dataset $x_1, \ldots, x_n$, the goal of offline MOO is to find Pareto-optimal solutions, i.e., candidates $x^\star \in \mathcal{X}$ for which there is no other $x \in \mathcal{X}$ with $f_i(x) \leq f_i(x^\star)$ for all $i \in \{1, \ldots, m\}$ and $f_j(x) < f_j(x^\star)$ for some $j \in \{1, \ldots, m\}$.

In this work, we follow the standard assumptions that (a) the offline dataset $x_1, \ldots, x_n$ is an independently and identically distributed (i.i.d.) sample from a probability measure $Q_{\text{off}}$ on $\mathcal{X}$, and (b) the Pareto front (objective values of Pareto-optimal solutions) can be viewed as a manifold $\mathcal{M}$.

**Generative method:** The goal of a generative method is to generate i.i.d. samples from a distribution $Q_{\text{alg}}$ that is close to the distribution $Q_{\text{off}}$ of a given dataset $x_1, \ldots, x_n$.

**Problem of this work:** We aim to diagnose possible improvements and distributional limitations of recent generative methods for offline MOO.

*Remark* 1 (on methodological approach). Note that generative methods are developed for generating samples from $Q_{\text{alg}}$ close to $Q_{\text{off}}$, while their application to offline MOO requires identifying samples possibly far from the pushforward distribution $P_{\text{off}} := \mathbf{f}_*(Q_{\text{off}})$ in the objective space with respect to the objective mapping $\mathbf{f}(x) := (f_1(x), \ldots, f_m(x))$, see Figure 1.

## 3 RELATED WORK

A common approach to offline optimization is to train a surrogate model on offline data and optimize it as a proxy for the unknown objective function (Xue et al., 2024; Kim et al., 2025). In single-objective settings, surrogates enable gradient-based optimization (Trabucco et al., 2021; Qi et al., 2022; Yuan et al., 2023; Yu et al., 2021; Chen et al., 2023), while in evolutionary computation they were first introduced for single-objective problems (Jones et al., 1998) and later extended to

multi-objective evolutionary algorithms (MOEAs) (Knowles, 2006; Jin, 2011; Wang et al., 2018; Yang et al., 2019). Recent generative modeling approaches have shown strong promise for offline optimization (Kumar & Levine, 2020; Krishnamoorthy et al., 2023; Mashkaria et al., 2023), but have largely focused on single-objective problems. Only limited work addresses offline multi-objective optimization, where flow- and diffusion-based models are used to guide sampling toward Pareto-optimal regions (Yuan et al., 2025; Annadani et al., 2025).

The offline setting is particularly relevant in real-world applications where function evaluations are expensive or time-consuming, such as engineering design (Tanabe & Ishibuchi, 2020). In this context, MOEAs are standard tools for electrical machine design in electrification and e-mobility (Chasparis et al., 2016; Bramerdorfer et al., 2018; Cavagnino et al., 2018). These problems are challenging due to the coupling of multiple physical domains, including electromagnetics, thermal effects, structural integrity, acoustics, and manufacturing constraints, resulting in complex objective landscapes that are typically evaluated via computationally intensive finite-element simulations (Silber et al., 2018; Bramerdorfer et al., 2016) (see Appendix A.1 for details).

## 4 THE OFFLINE-FRONTIER SHIFT

We quantify the offline-frontier shift using the squared expected projection distance (Hastie & Stuetzle, 1989, projection geometry) between the objective space offline distribution $P_{\text{off}}$ and the Pareto front $\mathcal{M}$:

$$s(P_{\text{off}}) := \mathbb{E}_{Y \sim P_{\text{off}}}\Big[\|Y - \Pi_{\mathcal{M}}(Y)\|_2^2\Big], \tag{1}$$

where $\Pi_{\mathcal{M}}$ denotes the orthogonal projection onto $\mathcal{M}$.

Note that $s(P)$ is a natural choice for measuring the quality of a distribution $P$ in MOO, as it converges to the squared generational distance $\text{GD}^2(Y_n, Z) := \frac{1}{n} \sum_{i=1}^{n} \min_{z \in Z} \|y_i - z\|_2^2$ (Schütze et al., 2012), where $Z$ is a finite discretization of $\mathcal{M}$; see Appendix A.3 for details. GD is well-known in evolutionary computation (Lamont & van Veldhuizen, 1999; Schütze et al., 2012).

**Lemma 1.** *The objective space offline distribution $P_{\text{off}}$ and the generated distribution $P_{\text{alg}}$ satisfy*

$$s(P_{\text{alg}}) \geq s(P_{\text{off}}) - d_{\mathcal{F}}(P_{\text{alg}}, P_{\text{off}}) \tag{2}$$

*for any integral probability metric $d_{\mathcal{F}}$ generated by a function class $\mathcal{F}$ that contains the measurable test function $\ell_{\mathcal{M}}(y) := \|y - \Pi_{\mathcal{M}}(y)\|_2^2$.*

*Proof.* By the triangle inequality, we obtain

$$
\begin{aligned}
s(P_{\text{off}}) &= s(P_{\text{off}}) + s(P_{\text{alg}}) - s(P_{\text{alg}}) \\
&\leq s(P_{\text{alg}}) + |s(P_{\text{off}}) - s(P_{\text{alg}})| \\
&= s(P_{\text{alg}}) + \Big|\mathbb{E}_{Y \sim P_{\text{off}}}\Big[\ell_{\mathcal{M}}(Y)\Big] - \mathbb{E}_{Y \sim P_{\text{alg}}}\Big[\ell_{\mathcal{M}}(Y)\Big]\Big| \\
&\leq s(P_{\text{alg}}) + \sup_{f \in \mathcal{F}}\Big|\mathbb{E}_{Y \sim P_{\text{off}}}\Big[f(Y)\Big] - \mathbb{E}_{Y \sim P_{\text{alg}}}\Big[f(Y)\Big]\Big| \\
&= s(P_{\text{alg}}) + d_{\mathcal{F}}(P_{\text{off}}, P_{\text{alg}}).
\end{aligned}
$$

Subtracting $d_{\mathcal{F}}(P_{\text{off}}, P_{\text{alg}})$ from both sides gives the desired result. □

Lemma 1 shows that

1. the performance of any classical generative MOO method trained to match $P_{\text{off}}$ is fundamentally limited by the offline-frontier shift $s(P_{\text{off}})$ alone;

2. performance improvements upon the offline distribution in MOO necessarily require out-of-distribution sampling, as quantified by the distance $d_{\mathcal{F}}(P_{\text{alg}}, P_{\text{off}})$ between the generated and offline distributions;

3. the Maximum Mean Discrepancy (Gretton et al., 2012), i.e., integral probability metric induced by the unit ball $\mathcal{F}$ of a reproducing kernel Hilbert space, is a natural measure of a generative method's improvement relative to the offline dataset.

## 5 EXPERIMENTS

We perform comprehensive experiments on a standard benchmark for offline MOO, Off-MOO-Bench (Xue et al., 2024). The goal of our experimental evaluation is to characterize the performance of generative methods across different MOO metrics, and to demonstrate that performance differences become more pronounced as the offline-frontier shift increases. We compare generative methods, including flow matching (Yuan et al., 2025) and diffusion (Annadani et al., 2025), to evolutionary methods. The evolutionary methods first train a surrogate model for the objective functions (Trabucco et al., 2021; Qi et al., 2022; Yuan et al., 2023; Yu et al., 2021; Chen et al., 2023; Yu et al., 2020; Chen et al., 2018) and then use NSGA-II (Deb et al., 2002) to search the design space. Additional training details are provided in Appendix A.5.

### 5.1 NOVEL MOO METRICS ON OFF-MOO-BENCH (XUE ET AL., 2024)

Table 1: MOO metrics on selected ZDT and DTLZ tasks. Results are reported for the best-performing evolutionary (Evo.) and generative (Gen.) methods. Each algorithm is run for 5 seeds and evaluated on 256 designs. Boldface indicates the best value.

|  |  | ZDT1 | ZDT2 | ZDT3 | DTLZ1 | DTLZ2 | DTLZ3 | DZLZ4 | DTLZ5 |
|---|---|---|---|---|---|---|---|---|---|
| HV ($\uparrow$) | Evo. | **4.83 ± 0.00** | 5.57 ± 0.05 | 5.59 ± 0.06 | **10.64 ± 0.00** | **12.44 ± 0.00** | **9.89 ± 0.00** | **17.70 ± 0.01** | **10.76 ± 0.00** |
|  | Gen. | 4.53 ± 0.03 | **5.67 ± 0.19** | **5.60 ± 0.05** | **10.64 ± 0.00** | 12.39 ± 0.01 | **9.89 ± 0.00** | 17.60 ± 0.03 | 10.59 ± 0.02 |
| GD+ ($\downarrow$) | Evo. | **0.01 ± 0.00** | **0.05 ± 0.03** | **0.17 ± 0.02** | **0.25 ± 0.04** | **0.02 ± 0.01** | **0.38 ± 0.16** | **0.18 ± 0.04** | **0.00 ± 0.00** |
|  | Gen. | 0.29 ± 0.03 | 0.18 ± 0.02 | 0.24 ± 0.02 | 0.40 ± 0.01 | 0.16 ± 0.03 | 0.52 ± 0.00 | 0.22 ± 0.00 | 0.16 ± 0.01 |
| IGD+ ($\downarrow$) | Evo. | **0.01 ± 0.00** | **0.04 ± 0.01** | 0.12 ± 0.01 | **0.08 ± 0.01** | **0.01 ± 0.00** | **0.09 ± 0.02** | **0.06 ± 0.01** | **0.00 ± 0.00** |
|  | Gen. | 0.13 ± 0.02 | 0.10 ± 0.02 | **0.10 ± 0.01** | 0.09 ± 0.03 | 0.12 ± 0.02 | 0.18 ± 0.02 | 0.20 ± 0.04 | 0.08 ± 0.01 |
| MMD ($\uparrow$) | Evo. | **1.02 ± 0.02** | **1.01 ± 0.04** | **0.95 ± 0.16** | **0.80 ± 0.06** | **0.41 ± 0.02** | **0.33 ± 0.41** | **0.37 ± 0.02** | **0.59 ± 0.01** |
|  | Gen. | 0.71 ± 0.08 | 0.86 ± 0.05 | 0.92 ± 0.06 | 0.54 ± 0.02 | 0.02 ± 0.05 | 0.00 ± 0.00 | 0.04 ± 0.08 | 0.23 ± 0.13 |

We evaluate the performance of a solution set using established MOO metrics: hypervolume (HV) (Zitzler & Thiele, 1999), generational distance plus (GD+) (Lamont & van Veldhuizen, 1999; Ishibuchi et al., 2015), and inverted generational distance plus (IGD+) (Coello Coello & Reyes Sierra, 2004; Ishibuchi et al., 2015). Formal definitions are provided in Appendix A.2. We report results for both evolutionary and generative methods on the ZDT and DTLZ tasks (Table 1). While generative methods perform competitively with evolutionary methods in terms of HV, they systematically underperform with respect to both GD+ and IGD+. We further show that generative methods behave more conservatively in objective space, remaining closer to the empirical offline distribution $\hat{P}_{\text{off}}$, as quantified by $\text{MMD}(\hat{P}_{\text{alg}}, \hat{P}_{\text{off}})$ (Figure 2). Moreover, we observe a correlation between distributional distance (MMD) and GD error, indicating that performance improvements necessarily require out-of-distribution sampling. Detailed results for HV, GD+, IGD+, and MMD are provided in Appendix A.6.

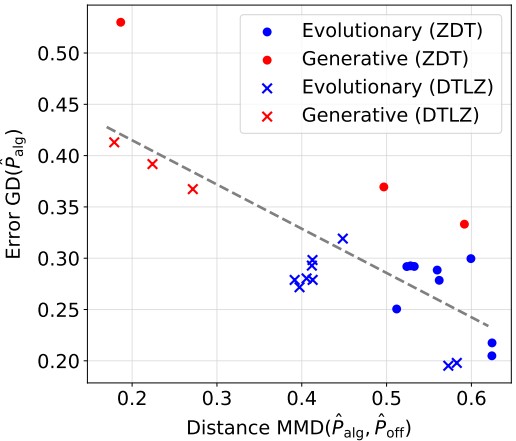

Figure 2: Correlation of error and distributional distance (MMD), averaged over subtasks.

### 5.2 OFFLINE-FRONTIER SHIFT AS AN INHERENT LIMITATION

We empirically demonstrate that increasing the offline-frontier shift degrades performance, with the effect significantly more pronounced for generative methods, as shown in Figure 1 and Table 2. To induce progressively larger shifts between the offline dataset and the Pareto front, we apply non-dominated sorting and iteratively remove 10,000 samples, starting from the first non-dominated

fronts. For each shift level $k$, we then randomly sample a new offline dataset of 10,000 points from the remaining data. Table 2 reports the resulting offline-frontier shifts $s(\hat{P}^k_{\text{off}})$ alongside the corresponding performance metrics.

Table 2: Offline-frontier shifts on modified ZDT tasks alongside MMD, error and error increase. Results are averaged over evolutionary (Evo.) and generative (Gen.) methods. Each algorithm is run for 5 seeds and evaluated on 256 designs. Boldface indicates the best value.

| | $s(\hat{P}^k_{\text{off}})$ | 0.461 | 0.486 | 0.513 | 0.545 | 0.585 | 0.632 |
|---|---|---|---|---|---|---|---|
| MMD ($\uparrow$) | Evo. | **0.67 ± 0.00** | **0.70 ± 0.00** | **0.84 ± 0.00** | **0.83 ± 0.00** | **0.85 ± 0.00** | **0.94 ± 0.00** |
| | Gen. | **0.67 ± 0.08** | 0.65 ± 0.07 | 0.74 ± 0.08 | 0.67 ± 0.04 | 0.65 ± 0.07 | 0.59 ± 0.02 |
| GD+ ($\downarrow$) | Evo. | **0.11 ± 0.00** | **0.11 ± 0.00** | **0.12 ± 0.00** | **0.12 ± 0.00** | **0.15 ± 0.00** | **0.17 ± 0.00** |
| | Gen. | 0.33 ± 0.01 | 0.34 ± 0.01 | 0.35 ± 0.01 | 0.37 ± 0.01 | 0.40 ± 0.02 | 0.46 ± 0.07 |
| $\Delta$GD+ ($\downarrow$) | Evo. | **0.00** | **0.00** | **0.01** | **0.01** | **0.04** | **0.06** |
| | Gen. | **0.00** | 0.01 | 0.02 | 0.04 | 0.07 | 0.13 |

# 6 DISCUSSION

Our results highlight limitations of current generative methods for offline MOO. While these models achieve strong hypervolume performance, they remain conservative in objective space, as measured by MMD, and have less ability to deviate into out-of-distribution regions compared to evolutionary alternatives. We find that this effect becomes more pronounced as the offline-frontier shift increases. This behavior aligns with the training objectives of generative models, remaining close to the offline data distribution. Consequently, a mismatch emerges between existing generative optimization methods and the requirements of offline MOO, motivating future work on methods that enable controlled extrapolation in objective space.

ACKNOWLEDGMENTS

The ELLIS Unit Linz, the LIT AI Lab, the Institute for Machine Learning, are supported by the Federal State Upper Austria. We thank the projects LCM-COMET K2 Center, FWF AIRI FG 9-N (10.55776/FG9), AI4GreenHeatingGrids (FFG-899943), ELISE (H2020-ICT-2019-3 ID: 951847), Stars4Waters (HORIZON-CL6-2021-CLIMATE-01-01), FWF Bilateral Artificial Intelligence ([10.55776/COE12]). We thank NXAI GmbH, Silicon Austria Labs (SAL), FILL Gesellschaft mbH, Google, ZF Friedrichshafen AG, Robert Bosch GmbH, Merck Healthcare KGaA, GLS (Univ. Waterloo), Borealis AG, TÜV Austria, TRUMPF and the NVIDIA Corporation.

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

# A APPENDIX

## APPENDIX CONTENTS

## A.1 REAL-WORLD APPLICATION

Electrical machine design is challenging due to the coupling of multiple physical domains, including electromagnetics, thermal effects, structural integrity, acoustics, and manufacturing constraints, resulting in complex objective landscapes that are typically evaluated via computationally intensive finite-element simulations (Silber et al., 2018; Bramerdorfer et al., 2016). The resulting structural complexity is illustrated in Figure 3, which shows a typical cross-section of a permanent-magnet synchronous machine for electric-vehicle applications. Traction-motor MOO problems commonly involve roughly 15–25 design variables, spanning geometric parameters (e.g., slot, tooth, and yoke dimensions, as well as magnet size and placement) and material selections (e.g., magnet grade and electrical steel). Objectives typically balance electromagnetic losses and efficiency against thermal limits (e.g., maximum winding or magnet temperatures) and noise, vibration, and harshness (NVH). These objectives are often aggregated over representative driving profiles such as the Worldwide harmonized Light vehicles Test Procedure (WLTP) cycle, for example by minimizing the total energy consumption or the cycle-integrated losses. Structural constraints include limits on permanent plastic deformation of the rotor core at burst speed.

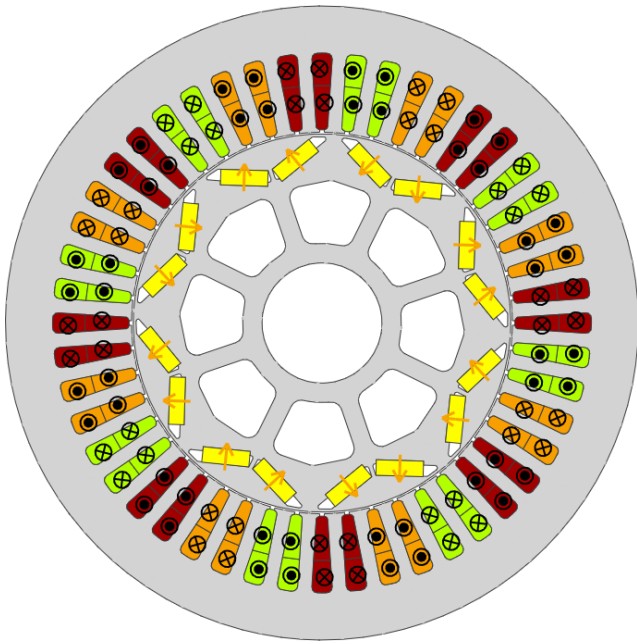

Figure 3: Cross-section of a permanent-magnet synchronous machine for electric-vehicles.

## A.2 MOO METRICS

Hypervolume (Zitzler & Thiele, 1999) measures the volume of the objective space dominated by the solution set $A \subset \mathbb{R}^m$ with respect to a reference point $r \in \mathbb{R}^m$:

$$\text{HV}(A, r) := \lambda\left( \bigcup_{a \in A} [a_1, r_1] \times [a_2, r_2] \times \cdots \times [a_m, r_m] \right), \tag{3}$$

where $\lambda$ denotes the Lebesgue measure in $\mathbb{R}^m$.

Generational distance (GD) (Lamont & van Veldhuizen, 1999; Schütze et al., 2012) is computed as the average distance from a point in the solution set $A$ to its closest point in the Pareto front $Z$:

$$\text{GD}(A, Z) := \sqrt{\frac{1}{|A|} \sum_{a \in A} d(a, Z)^2}, \tag{4}$$

where $d(x, Y) := \min_{y \in Y} \|x - y\|_2$ denotes the distance between point $x$ and set $Y$.

Analogously, the inverted generational distance (IGD) (Coello Coello & Reyes Sierra, 2004; Schütze et al., 2012) is computed as the average distance from a point in the Pareto front $Z$ to its closest point in the solution set $A$:

$$\text{IGD}(A, Z) := \text{GD}(Z, A) = \sqrt{\frac{1}{|Z|} \sum_{z \in Z} d(z, A)^2}. \tag{5}$$

Ishibuchi et al. (2015) propose the following modification of the standard GD and IGD:

$$d^+(x, Y) := \min_{y \in Y} \Big\| \max\{h(x, y), 0\} \Big\|_2,$$

where

$$h(x, y) := \begin{cases} x - y & \text{for } \text{GD}^+, \\ y - x & \text{for } \text{IGD}^+. \end{cases}$$

The maximum operator is applied component-wise and ensures that only positive deviations, corresponding to improvements along each objective, are considered in the distance computation.

## A.3 OFFLINE-FRONTIER SHIFT AS A GENERALIZATION OF THE GD

The offline-frontier shift $s$ is a natural generalization of GD: it replaces finite subsets $Z$ with (piecewise smooth) manifolds $\mathcal{M}$, discrete minimum distances $d(\cdot, Z)$ with orthogonal projections $\Pi_{\mathcal{M}}$, and finite averages with expectations over probability distributions. We show that the squared GD of the offline dataset converges to the offline-frontier shift evaluated at the associated empirical distribution.

The offline dataset in objective space $Y := \{y_1, \ldots, y_n\}$ with $y_i := \mathbf{f}(x_i)$ is associated with the empirical distribution

$$\hat{P}_{\text{off}} := \frac{1}{n} \sum_{i=1}^{n} \delta_{y_i},$$

and therefore,

$$s(\hat{P}_{\text{off}}) = \mathbb{E}_{Y \sim \hat{P}_{\text{off}}}\left[ \|Y - \Pi_{\mathcal{M}}(Y)\|_2^2 \right] = \frac{1}{n} \sum_{i=1}^{n} \|y_i - \Pi_{\mathcal{M}}(y_i)\|_2^2.$$

In evolutionary MOO, the Hausdorff distance (Schütze et al., 2012) provides a classical measure of distance between the Pareto front and its approximation. We extend this notion to analyze how well finite discretizations approximate the true Pareto manifold. In particular, we establish that the squared GD converges to the offline-frontier shift evaluated at the associated empirical distribution as the discretizations converge to the Pareto manifold in the Hausdorff sense (Lemma 2).

The Hausdorff distance $d_{\mathcal{H}}$ between two sets $A, B \subset \mathbb{R}^m$ is defined as

$$d_{\mathcal{H}}(A, B) := \max\Big\{ \sup_{a \in A} \inf_{b \in B} \|a - b\|_2, \ \sup_{b \in B} \inf_{a \in A} \|b - a\|_2 \Big\}. \tag{6}$$

*Remark* 2. A sequence of sets $(Z_k)_{k \in \mathbb{N}}$, $Z_k \subset \mathcal{M}$, converges to $\mathcal{M}$ in Hausdorff distance if and only if $d_{\mathcal{H}}(Z_k, \mathcal{M}) = \sup_{y \in \mathcal{M}} \inf_{z \in Z_k} \|y - z\| \longrightarrow 0$ as $k \to \infty$.

**Lemma 2** (Convergence of GD to expected projection distance). *Let $\mathcal{M} \subset \mathbb{R}^m$ be compact, and let $P$ be a probability measure over $\mathbb{R}^m$ such that*

$$s(P) = \mathbb{E}_{X \sim P}[\|X - \Pi_{\mathcal{M}}(X)\|_2^2] < \infty,$$

*where $\Pi_{\mathcal{M}}(x) := \arg\min_{y \in \mathcal{M}} \|x - y\|_2$. Let $Y_n = \{y_1, \ldots, y_n\}$ be i.i.d. samples from P, and let $(Z_k)_{k \in \mathbb{N}}$ be a sequence of finite subsets $Z_k \subset \mathcal{M}$ with*

$$d_{\mathcal{H}}(Z_k, \mathcal{M}) \to 0 \quad as\ k \to \infty.$$

*Then, almost surely,*

$$\lim_{n \to \infty, k \to \infty} \mathrm{GD}^2(Y_n, Z_k) = s(P).$$

*Proof.* We prove the convergence by splitting the error into two parts:

**a) Discretization error:** Fix a sample $y_i \in Y_n$. For each $k$, define

$$z_k^i := \arg\min_{z \in Z_k} \|y_i - z\|_2.$$

By compactness of $\mathcal{M}$, the projection

$$\Pi_{\mathcal{M}}(y_i) := \arg\min_{y \in \mathcal{M}} \|y_i - y\|_2$$

exists. Fix $\varepsilon > 0$. By Hausdorff convergence, there exists $k' \in \mathbb{N}$ such that for all $k \geq k'$,

$$\inf_{z \in Z_k} \|\Pi_{\mathcal{M}}(y_i) - z\|_2 \leq \varepsilon.$$

Hence, for such $k$, there exists $z_k' \in Z_k$ with

$$\|\Pi_{\mathcal{M}}(y_i) - z_k'\|_2 \leq \varepsilon.$$

By minimality of $z_k^i$ and triangle inequality,

$$\|y_i - z_k^i\|_2 \leq \|y_i - z_k'\|_2 \leq \|y_i - \Pi_{\mathcal{M}}(y_i)\|_2 + \|\Pi_{\mathcal{M}}(y_i) - z_k'\|_2 \leq \|y_i - \Pi_{\mathcal{M}}(y_i)\|_2 + \varepsilon.$$

On the other hand, since $z_k^i \in Z_k \subset \mathcal{M}$,

$$\|y_i - \Pi_{\mathcal{M}}(y_i)\|_2 \leq \|y_i - z_k^i\|_2.$$

Combining the inequalities gives

$$\|y_i - \Pi_{\mathcal{M}}(y_i)\|_2 \leq \|y_i - z_k^i\|_2 \leq \|y_i - \Pi_{\mathcal{M}}(y_i)\|_2 + \varepsilon.$$

Since $\varepsilon > 0$ was arbitrary, taking the limit $\epsilon \to 0$ gives

$$\lim_{k \to \infty} \|y_i - z_i^k\|_2 = \|y_i - \Pi_{\mathcal{M}}(y_i)\|_2.$$

Applying this point-wise convergence to all $y_i \in Y_n$ immediately gives

$$\lim_{k \to \infty} \mathrm{GD}^2(Y_n, Z_k) = \frac{1}{n} \sum_{i=1}^{n} \|y_i - \Pi_{\mathcal{M}}(y_i)\|_2^2 = s(\hat{P}_n),$$

i.e., $\mathrm{GD}^2(Y_n, \cdot)$ converges to the empirical squared projection distance.

**b) Sampling error:** Define

$$X_i := \|y_i - \Pi_{\mathcal{M}}(y_i)\|_2^2.$$

Then $X_1, \ldots, X_n$ are i.i.d. with $\mathbb{E}[X_i] = s(P) < \infty$. By the strong law of large numbers (SLLN),

$$s(\hat{P}_n) = \frac{1}{n} \sum_{i=1}^{n} X_i \xrightarrow{\text{a.s.}} s(P) \quad as\ n \to \infty.$$

**c) Combination:** Finally, by the triangle inequality,

$$|\mathrm{GD}^2(Y_n, Z_k) - s(P)| \leq |\mathrm{GD}^2(Y_n, Z_k) - s(\hat{P}_n)| + |s(\hat{P}_n) - s(P)|.$$

The first term vanishes as $k \to \infty$ (discretization error). The second term vanishes as $n \to \infty$ (sampling error). Hence, taking the joint limit $n \to \infty$, $k \to \infty$, we get

$$\mathrm{GD}^2(Y_n, Z_k) \xrightarrow{\text{a.s.}} s(P) \quad as\ n \to \infty, k \to \infty.$$

$\square$

## A.4 Integral Probability Metrics

Given two probability measures $P$ and $Q$ on a measurable space $\mathcal{Y}$ and a class of measurable functions $\mathcal{F}$, the induced integral probability metric (IPM) $d_{\mathcal{F}}$ is defined as

$$d_{\mathcal{F}}(P, Q) := \sup_{f \in \mathcal{F}} \left| \mathbb{E}_{Y \sim P} \left[ f(Y) \right] - \mathbb{E}_{Y \sim Q} \left[ f(Y) \right] \right|. \tag{7}$$

The IPM $d_{\mathcal{F}}$ induced by the unit ball $\mathcal{F}$ of a reproducing kernel Hilbert space is the Maximum Mean Discrepency (MMD) (Gretton et al., 2012). With associated kernel $k(\cdot, \cdot)$, the squared MMD is

$$\text{MMD}^2(P, Q) = \mathbb{E}_{x, x' \sim P} \left[ k(x, x') \right] + \mathbb{E}_{y, y' \sim Q} \left[ k(y, y') \right] - 2 \mathbb{E}_{x \sim P, y \sim Q} \left[ k(x, y) \right]. \tag{8}$$

Similar to the modification in the GD+ and IGD+, we only consider positive deviations in the distance computation $\text{MMD}(\hat{P}_{\text{alg}}, \hat{P}_{\text{off}})$.

## A.5 Training Details

Following the approach of (Xue et al., 2024), we consider two DNN-based architecture models: multi-head and multiple-models. We train 1) multi-head models, including multi-head vanilla, multi-head + GradNorm (Chen et al., 2018), and multi-head + PcGrad (Yu et al., 2020), and 2) multiple-models, including multiple-models vanilla, multiple-models + COM (Trabucco et al., 2021), multiple-models + IOM (Qi et al., 2022), multiple-models + RoMA (Yu et al., 2021), multiple-models + ICT (Yuan et al., 2023), and multiple-models + TriMentoring (Chen et al., 2023).

The multi-head model consists of a shared MLP feature extractor with two 2048-dimensional hidden layers and ReLU activations, followed by one task-specific head per objective (one 2048-dimensional hidden layer and ReLU activation). Multiple-models trains an independent MLP for each objective with the same architecture as the multi-head model. All models are trained on the offline dataset using mean squared error (MSE) loss and the Adam optimizer with initial learning rate $1 \times 10^{-3}$ and learning rate decay $0.98$ per epoch. Training proceeds for 200 epochs with batch size 128.

The flow matching model (Yuan et al., 2025) uses a 4-layer MLP with SeLU activations and hidden layer size of $512$. Training runs for 1000 epochs with early stopping and Adam optimizer. Following the approach of Annadani et al. (2025), we train a preference model and an unconditional diffusion model. The unconditional diffusion model is an MLP with two 512-dimensional hidden layers and sinusoidal time embedding, followed by ReLU activation and layer normalization. The model is trained using AdamW with learning rate $5 \times 10^{-4}$ for up to 200 epochs. The preference model is an MLP with three hidden layers: the first two hidden layers match the input dimensionality, while the last hidden layer has $512$ units. ReLU nonlinearity, layer normalization, and sinusoidal time embeddings are applied similarly to the denoising model. The model is trained using Adam with learning rate $1 \times 10^{-5}$ for up to $500$ epochs.

## A.6 ADDITIONAL EXPERIMENTAL RESULTS

Table 3: HV results of ZDT tasks (↑). Each algorithm is run for 5 seeds and evaluated on 256 designs. Bold numbers indicate the best performance.

|  | ZDT1 | ZDT2 | ZDT3 | ZDT4 | ZDT6 | Avg. Rank |
|---|---|---|---|---|---|---|
| D(best) | 4.17 ± 0.00 | 4.68 ± 0.00 | 5.15 ± 0.00 | 5.46 ± 0.00 | 4.61 ± 0.00 | 10.0 ± 5.2 |
| MultiHead-Vallina | 4.80 ± 0.05 | 5.57 ± 0.09 | 5.59 ± 0.06 | 4.74 ± 0.31 | 4.78 ± 0.00 | **4.2 ± 1.92** |
| MultiHead-PcGrad | **4.83 ± 0.03** | 5.56 ± 0.09 | 5.51 ± 0.03 | 4.25 ± 0.46 | 4.77 ± 0.02 | 5.6 ± 1.14 |
| MultiHead-GradNorm | 4.69 ± 0.16 | 5.38 ± 0.14 | 5.49 ± 0.21 | 3.95 ± 0.40 | 4.30 ± 0.83 | 10.4 ± 1.95 |
| MultipleModels-Vallina | 4.80 ± 0.05 | 5.57 ± 0.09 | 5.59 ± 0.06 | 4.74 ± 0.31 | 4.78 ± 0.00 | **4.2 ± 1.92** |
| MultipleModels-COM | **4.83 ± 0.00** | 5.54 ± 0.07 | 5.43 ± 0.03 | 4.09 ± 0.20 | 4.73 ± 0.05 | 6.4 ± 3.13 |
| MultipleModels-IOM | **4.83 ± 0.01** | 5.57 ± 0.05 | 5.51 ± 0.05 | 3.91 ± 0.27 | 4.75 ± 0.01 | 5.8 ± 3.77 |
| MultipleModels-RoMA | **4.83 ± 0.01** | 5.57 ± 0.05 | 5.51 ± 0.05 | 3.91 ± 0.27 | 4.75 ± 0.01 | 5.8 ± 3.77 |
| MultipleModels-ICT | **4.83 ± 0.00** | 5.54 ± 0.07 | 5.43 ± 0.03 | 4.09 ± 0.20 | 4.73 ± 0.05 | 6.4 ± 3.13 |
| MultipleModels-TriMentoring | **4.83 ± 0.00** | 5.54 ± 0.07 | 5.43 ± 0.03 | 4.09 ± 0.20 | 4.73 ± 0.05 | 6.4 ± 3.13 |
| ParetoFlow-Vallina | 4.19 ± 0.06 | **5.67 ± 0.19** | 5.18 ± 0.07 | 4.91 ± 0.14 | 4.53 ± 0.06 | 8.0 ± 5.15 |
| Diffusion-Guidance-Crowding | 4.53 ± 0.03 | 5.39 ± 0.04 | **5.60 ± 0.05** | 5.01 ± 0.08 | **4.82 ± 0.01** | 5.0 ± 4.64 |
| Diffusion-Guidance-SubCrowding | 4.53 ± 0.05 | 5.25 ± 0.04 | **5.60 ± 0.05** | **5.03 ± 0.06** | 4.49 ± 0.19 | 7.6 ± 5.59 |

Table 4: HV results of DTLZ tasks (↑). Each algorithm is run for 5 seeds and evaluated on 256 designs. Bold numbers indicate the best performance.

|  | DTLZ1 | DTLZ2 | DTLZ3 | DTLZ4 | DTLZ5 | DTLZ6 | DTLZ7 | Avg. Rank |
|---|---|---|---|---|---|---|---|---|
| D(best) | 10.63 ± 0.00 | 12.43 ± 0.00 | 9.90 ± 0.00 | 17.50 ± 0.00 | 10.69 ± 0.00 | 10.69 ± 0.00 | 9.04 ± 0.00 | 9.0 ± 3.79 |
| MultiHead-Vallina | 10.56 ± 0.16 | **12.44 ± 0.00** | 9.75 ± 0.27 | **17.70 ± 0.01** | **10.76 ± 0.00** | **10.95 ± 0.01** | 10.60 ± 0.02 | 4.86 ± 5.05 |
| MultiHead-PcGrad | **10.64 ± 0.00** | **12.44 ± 0.00** | **9.89 ± 0.00** | 17.66 ± 0.02 | **10.76 ± 0.00** | 10.93 ± 0.01 | 10.62 ± 0.03 | 3.14 ± 2.79 |
| MultiHead-GradNorm | 10.57 ± 0.15 | 11.90 ± 1.08 | 9.66 ± 0.28 | 17.64 ± 0.04 | 10.01 ± 1.17 | 10.79 ± 0.09 | 9.79 ± 0.71 | 11.14 ± 1.86 |
| MultipleModels-Vallina | 10.56 ± 0.16 | **12.44 ± 0.00** | 9.75 ± 0.27 | **17.70 ± 0.01** | **10.76 ± 0.00** | **10.95 ± 0.01** | 10.60 ± 0.02 | 4.86 ± 5.05 |
| MultipleModels-COM | **10.64 ± 0.00** | **12.44 ± 0.00** | **9.89 ± 0.00** | 17.69 ± 0.02 | **10.76 ± 0.00** | 10.90 ± 0.01 | **10.74 ± 0.01** | **1.86 ± 1.21** |
| MultipleModels-IOM | **10.64 ± 0.00** | **12.44 ± 0.00** | **9.89 ± 0.00** | 17.68 ± 0.01 | **10.76 ± 0.00** | 10.89 ± 0.01 | **10.74 ± 0.03** | 3.14 ± 2.54 |
| MultipleModels-RoMA | **10.64 ± 0.00** | **12.44 ± 0.00** | **9.89 ± 0.00** | 17.68 ± 0.01 | **10.76 ± 0.00** | 10.89 ± 0.01 | **10.74 ± 0.03** | 3.14 ± 2.54 |
| MultipleModels-ICT | **10.64 ± 0.00** | **12.44 ± 0.00** | **9.89 ± 0.00** | 17.69 ± 0.02 | **10.76 ± 0.00** | 10.90 ± 0.01 | **10.74 ± 0.01** | **1.86 ± 1.21** |
| MultipleModels-TriMentoring | **10.64 ± 0.00** | **12.44 ± 0.00** | **9.89 ± 0.00** | 17.69 ± 0.02 | **10.76 ± 0.00** | 10.90 ± 0.01 | **10.74 ± 0.01** | **1.86 ± 1.21** |
| ParetoFlow-Vallina | 10.60 ± 0.02 | 12.27 ± 0.04 | 9.82 ± 0.04 | 15.93 ± 0.01 | 10.59 ± 0.02 | 10.71 ± 0.05 | 8.87 ± 0.09 | 11.14 ± 1.21 |
| Diffusion-Guidance-Crowding | **10.64 ± 0.00** | 12.39 ± 0.01 | **9.89 ± 0.00** | 15.02 ± 0.98 | 10.55 ± 0.03 | 10.53 ± 0.09 | 9.78 ± 0.08 | 8.71 ± 5.09 |
| Diffusion-Guidance-SubCrowding | **10.64 ± 0.00** | 12.38 ± 0.01 | 9.88 ± 0.00 | 17.60 ± 0.03 | 10.57 ± 0.02 | 10.80 ± 0.03 | 9.59 ± 0.12 | 8.86 ± 3.58 |

Table 5: GD+ results of ZDT tasks (↓). Each algorithm is run for 5 seeds and evaluated on 256 designs. Boldface indicates the best-performing methods.

|  |  | ZDT1 | ZDT2 | ZDT3 | ZDT4 | ZDT6 | Avg. |
|---|---|---|---|---|---|---|---|
|  | D(best) | 0.38 ± 0.00 | 0.43 ± 0.00 | 0.38 ± 0.00 | 0.05 ± 0.00 | 0.06 ± 0.00 | 0.26 ± 0.17 |
| Evo. | MultiHead-Vallina | 0.04 ± 0.02 | **0.05 ± 0.03** | **0.17 ± 0.02** | 0.68 ± 0.08 | 0.10 ± 0.00 | **0.21 ± 0.24** |
|  | MultiHead-PcGrad | **0.01 ± 0.00** | 0.08 ± 0.05 | 0.26 ± 0.01 | 0.79 ± 0.24 | 0.14 ± 0.15 | 0.26 ± 0.28 |
|  | MultiHead-GradNorm | 0.08 ± 0.05 | 0.15 ± 0.04 | **0.17 ± 0.02** | 0.96 ± 0.10 | 0.11 ± 0.05 | 0.29 ± 0.33 |
|  | MultipleModels-Vallina | 0.04 ± 0.02 | **0.05 ± 0.03** | **0.17 ± 0.02** | 0.68 ± 0.08 | 0.10 ± 0.00 | **0.21 ± 0.24** |
|  | MultipleModels-COM | 0.03 ± 0.01 | 0.10 ± 0.03 | 0.29 ± 0.01 | 0.90 ± 0.11 | 0.17 ± 0.17 | 0.30 ± 0.31 |
|  | MultipleModels-IOM | 0.02 ± 0.00 | 0.08 ± 0.01 | 0.25 ± 0.02 | 0.97 ± 0.10 | **0.08 ± 0.07** | 0.28 ± 0.35 |
|  | MultipleModels-RoMA | 0.02 ± 0.00 | 0.08 ± 0.01 | 0.25 ± 0.02 | 0.97 ± 0.10 | **0.08 ± 0.07** | 0.28 ± 0.35 |
|  | MultipleModels-ICT | 0.03 ± 0.01 | 0.10 ± 0.03 | 0.29 ± 0.01 | 0.90 ± 0.11 | 0.17 ± 0.17 | 0.30 ± 0.31 |
|  | MultipleModels-TriMentoring | 0.03 ± 0.01 | 0.10 ± 0.03 | 0.29 ± 0.01 | 0.90 ± 0.11 | 0.17 ± 0.17 | 0.30 ± 0.31 |
| Gen. | ParetoFlow-Vallina | 0.34 ± 0.02 | 0.44 ± 0.01 | 0.36 ± 0.02 | **0.39 ± 0.06** | 0.17 ± 0.00 | 0.34 ± 0.09 |
|  | Diffusion-Guidance-Crowding | 0.29 ± 0.03 | 0.18 ± 0.02 | 0.24 ± 0.02 | 0.56 ± 0.01 | 0.51 ± 0.04 | 0.36 ± 0.15 |
|  | Diffusion-Guidance-SubCrowding | 0.39 ± 0.03 | 0.48 ± 0.01 | 0.37 ± 0.02 | 0.56 ± 0.00 | 0.84 ± 0.02 | 0.53 ± 0.17 |

Table 6: GD+ results of DTLZ tasks (↓). Each algorithm is run for 5 seeds and evaluated on 256 designs. Boldface indicates the best-performing methods.

|  |  | DTLZ1 | DTLZ2 | DTLZ3 | DTLZ4 | DTLZ5 | DTLZ6 | DTLZ7 | Avg. |
|---|---|---|---|---|---|---|---|---|---|
|  | D(best) | 0.60 ± 0.00 | 0.10 ± 0.00 | 0.34 ± 0.00 | 0.18 ± 0.00 | 0.13 ± 0.00 | 0.76 ± 0.00 | 0.48 ± 0.00 | 0.37 ± 0.23 |
| Evo. | MultiHead-Vallina | **0.25 ± 0.04** | **0.02 ± 0.01** | 0.42 ± 0.03 | **0.18 ± 0.04** | **0.00 ± 0.00** | 0.46 ± 0.01 | 0.08 ± 0.07 | **0.20 ± 0.17** |
|  | MultiHead-PcGrad | 0.44 ± 0.03 | 0.07 ± 0.03 | 0.44 ± 0.03 | 0.30 ± 0.02 | 0.05 ± 0.00 | 0.51 ± 0.01 | **0.06 ± 0.02** | 0.27 ± 0.19 |
|  | MultiHead-GradNorm | 0.34 ± 0.11 | 0.25 ± 0.28 | **0.38 ± 0.16** | 0.24 ± 0.04 | 0.22 ± 0.20 | 0.59 ± 0.05 | 0.22 ± 0.09 | 0.32 ± 0.12 |
|  | MultipleModels-Vallina | **0.25 ± 0.04** | **0.02 ± 0.01** | 0.42 ± 0.03 | **0.18 ± 0.04** | **0.00 ± 0.00** | 0.46 ± 0.01 | 0.08 ± 0.07 | **0.20 ± 0.17** |
|  | MultipleModels-COM | 0.44 ± 0.04 | 0.06 ± 0.01 | 0.50 ± 0.04 | 0.27 ± 0.04 | 0.06 ± 0.02 | 0.55 ± 0.02 | 0.15 ± 0.01 | 0.29 ± 0.19 |
|  | MultipleModels-IOM | 0.44 ± 0.02 | 0.05 ± 0.01 | 0.47 ± 0.01 | 0.25 ± 0.03 | 0.06 ± 0.01 | 0.56 ± 0.02 | 0.14 ± 0.03 | 0.28 ± 0.19 |
|  | MultipleModels-RoMA | 0.44 ± 0.02 | 0.05 ± 0.01 | 0.47 ± 0.01 | 0.25 ± 0.03 | 0.06 ± 0.01 | 0.56 ± 0.02 | 0.14 ± 0.03 | 0.28 ± 0.19 |
|  | MultipleModels-ICT | 0.44 ± 0.04 | 0.06 ± 0.01 | 0.50 ± 0.04 | 0.27 ± 0.04 | 0.06 ± 0.02 | 0.55 ± 0.02 | 0.15 ± 0.01 | 0.29 ± 0.19 |
|  | MultipleModels-TriMentoring | 0.44 ± 0.04 | 0.06 ± 0.01 | 0.50 ± 0.04 | 0.27 ± 0.04 | 0.06 ± 0.02 | 0.55 ± 0.02 | 0.15 ± 0.01 | 0.29 ± 0.19 |
| Gen. | ParetoFlow-Vallina | 0.45 ± 0.02 | 0.16 ± 0.03 | 0.53 ± 0.04 | 0.22 ± 0.00 | 0.16 ± 0.01 | 0.75 ± 0.05 | 0.50 ± 0.03 | 0.40 ± 0.21 |
|  | Diffusion-Guidance-Crowding | 0.40 ± 0.01 | 0.22 ± 0.02 | 0.52 ± 0.00 | 0.22 ± 0.02 | 0.24 ± 0.04 | 0.60 ± 0.02 | 0.35 ± 0.02 | 0.36 ± 0.14 |
|  | Diffusion-Guidance-SubCrowding | 0.42 ± 0.01 | 0.27 ± 0.01 | 0.52 ± 0.01 | 0.25 ± 0.01 | 0.29 ± 0.00 | 0.73 ± 0.01 | 0.48 ± 0.01 | 0.42 ± 0.16 |

Table 7: IGD+ results of ZDT tasks (↓). Each algorithm is run for 5 seeds and evaluated on 256 designs. Boldface indicates the best-performing methods.

|  |  | ZDT1 | ZDT2 | ZDT3 | ZDT4 | ZDT6 | Avg. |
|---|---|---|---|---|---|---|---|
|  | D(best) | 0.36 ± 0.00 | 0.44 ± 0.00 | 0.36 ± 0.00 | 0.05 ± 0.00 | 0.07 ± 0.00 | 0.26 ± 0.16 |
| Evo. | MultiHead-Vallina | 0.02 ± 0.01 | 0.04 ± 0.02 | 0.12 ± 0.01 | 0.38 ± 0.14 | 0.02 ± 0.00 | **0.12 ± 0.14** |
|  | MultiHead-PcGrad | 0.01 ± 0.01 | 0.04 ± 0.02 | 0.20 ± 0.01 | 0.57 ± 0.22 | 0.02 ± 0.00 | 0.17 ± 0.21 |
|  | MultiHead-GradNorm | 0.08 ± 0.06 | 0.12 ± 0.05 | 0.15 ± 0.05 | 0.74 ± 0.18 | 0.13 ± 0.18 | 0.24 ± 0.25 |
|  | MultipleModels-Vallina | 0.02 ± 0.01 | 0.04 ± 0.02 | 0.12 ± 0.01 | 0.38 ± 0.14 | 0.02 ± 0.00 | **0.12 ± 0.14** |
|  | MultipleModels-COM | 0.02 ± 0.00 | 0.05 ± 0.02 | 0.23 ± 0.01 | 0.67 ± 0.09 | 0.04 ± 0.02 | 0.20 ± 0.25 |
|  | MultipleModels-IOM | **0.01 ± 0.00** | **0.04 ± 0.01** | 0.20 ± 0.02 | 0.76 ± 0.12 | 0.03 ± 0.01 | 0.21 ± 0.28 |
|  | MultipleModels-RoMA | **0.01 ± 0.00** | **0.04 ± 0.01** | 0.20 ± 0.02 | 0.76 ± 0.12 | 0.03 ± 0.01 | 0.21 ± 0.28 |
|  | MultipleModels-ICT | 0.02 ± 0.00 | 0.05 ± 0.02 | 0.23 ± 0.01 | 0.67 ± 0.09 | 0.04 ± 0.02 | 0.20 ± 0.25 |
|  | MultipleModels-TriMentoring | 0.02 ± 0.00 | 0.05 ± 0.02 | 0.23 ± 0.01 | 0.67 ± 0.09 | 0.04 ± 0.02 | 0.20 ± 0.25 |
| Gen. | ParetoFlow-Vallina | 0.32 ± 0.01 | 0.39 ± 0.02 | 0.33 ± 0.03 | 0.30 ± 0.06 | 0.10 ± 0.03 | 0.29 ± 0.10 |
|  | Diffusion-Guidance-Crowding | 0.13 ± 0.02 | 0.10 ± 0.02 | **0.10 ± 0.01** | 0.26 ± 0.04 | **0.01 ± 0.01** | **0.12 ± 0.08** |
|  | Diffusion-Guidance-SubCrowding | 0.14 ± 0.01 | 0.17 ± 0.02 | 0.11 ± 0.01 | **0.25 ± 0.04** | 0.22 ± 0.10 | 0.18 ± 0.05 |

Table 8: IGD+ results of DTLZ tasks (↓). Each algorithm is run for 5 seeds and evaluated on 256 designs. Boldface indicates the best-performing methods.

|  |  | DTLZ1 | DTLZ2 | DTLZ3 | DTLZ4 | DTLZ5 | DTLZ6 | DTLZ7 | Avg. |
|---|---|---|---|---|---|---|---|---|---|
|  | D(best) | 0.26 ± 0.00 | 0.05 ± 0.00 | 0.08 ± 0.00 | 0.15 ± 0.00 | 0.05 ± 0.00 | 0.68 ± 0.00 | 0.37 ± 0.00 | 0.23 ± 0.21 |
| Evo. | MultiHead-Vallina | 0.10 ± 0.01 | **0.01 ± 0.00** | 0.16 ± 0.02 | **0.06 ± 0.01** | **0.00 ± 0.00** | 0.40 ± 0.02 | **0.02 ± 0.00** | **0.11 ± 0.13** |
|  | MultiHead-PcGrad | 0.10 ± 0.03 | **0.01 ± 0.00** | **0.09 ± 0.02** | 0.10 ± 0.01 | **0.00 ± 0.00** | 0.45 ± 0.01 | 0.02 ± 0.01 | 0.11 ± 0.14 |
|  | MultiHead-GradNorm | **0.08 ± 0.01** | 0.18 ± 0.30 | 0.14 ± 0.02 | 0.11 ± 0.04 | 0.13 ± 0.22 | 0.51 ± 0.03 | 0.25 ± 0.10 | 0.20 ± 0.14 |
|  | MultipleModels-Vallina | 0.10 ± 0.01 | **0.01 ± 0.00** | 0.16 ± 0.02 | **0.06 ± 0.01** | **0.00 ± 0.00** | 0.40 ± 0.02 | **0.02 ± 0.00** | **0.11 ± 0.13** |
|  | MultipleModels-COM | 0.11 ± 0.01 | **0.01 ± 0.00** | 0.16 ± 0.04 | 0.08 ± 0.03 | **0.00 ± 0.00** | 0.47 ± 0.02 | **0.02 ± 0.00** | 0.12 ± 0.15 |
|  | MultipleModels-IOM | 0.12 ± 0.03 | 0.02 ± 0.00 | 0.17 ± 0.03 | 0.09 ± 0.02 | **0.00 ± 0.00** | 0.48 ± 0.02 | 0.02 ± 0.01 | 0.13 ± 0.15 |
|  | MultipleModels-RoMA | 0.12 ± 0.03 | 0.02 ± 0.00 | 0.17 ± 0.03 | 0.09 ± 0.02 | **0.00 ± 0.00** | 0.48 ± 0.02 | 0.02 ± 0.01 | 0.13 ± 0.15 |
|  | MultipleModels-ICT | 0.11 ± 0.01 | **0.01 ± 0.00** | 0.16 ± 0.04 | 0.08 ± 0.03 | **0.00 ± 0.00** | 0.47 ± 0.02 | **0.02 ± 0.00** | 0.12 ± 0.15 |
|  | MultipleModels-TriMentoring | 0.11 ± 0.01 | **0.01 ± 0.00** | 0.16 ± 0.04 | 0.08 ± 0.03 | **0.00 ± 0.00** | 0.47 ± 0.02 | **0.02 ± 0.00** | 0.12 ± 0.15 |
| Gen. | ParetoFlow-Vallina | 0.11 ± 0.05 | 0.12 ± 0.02 | 0.19 ± 0.05 | 0.29 ± 0.00 | 0.11 ± 0.01 | 0.56 ± 0.03 | 0.37 ± 0.01 | 0.25 ± 0.16 |
|  | Diffusion-Guidance-Crowding | 0.09 ± 0.03 | 0.17 ± 0.03 | 0.18 ± 0.02 | 0.30 ± 0.05 | 0.08 ± 0.01 | **0.24 ± 0.05** | 0.19 ± 0.02 | 0.18 ± 0.07 |
|  | Diffusion-Guidance-SubCrowding | 0.10 ± 0.03 | 0.12 ± 0.02 | 0.18 ± 0.05 | 0.20 ± 0.04 | 0.11 ± 0.02 | 0.42 ± 0.07 | 0.23 ± 0.03 | 0.19 ± 0.10 |

Table 9: $\mathrm{MMD}(\hat{P}_{\mathrm{alg}}, \hat{P}_{\mathrm{off}})$ results of ZDT tasks (↑). Each algorithm is run for 5 seeds and evaluated on 256 designs. Boldface marks the methods with the largest distance.

| | ZDT1 | ZDT2 | ZDT3 | ZDT4 | ZDT6 | Avg. |
|---|---|---|---|---|---|---|
| MultiHead-Vallina | 1.02 ± 0.10 | 0.97 ± 0.10 | 0.93 ± 0.11 | 0.16 ± 0.09 | 0.00 ± 0.00 | **0.62 ± 0.44** |
| MultiHead-PcGrad | 0.99 ± 0.06 | 0.95 ± 0.13 | 0.53 ± 0.05 | 0.08 ± 0.09 | 0.00 ± 0.00 | 0.51 ± 0.42 |
| MultiHead-GradNorm | 0.92 ± 0.11 | 0.87 ± 0.14 | **0.95 ± 0.16** | 0.08 ± 0.10 | 0.14 ± 0.26 | 0.59 ± 0.39 |
| MultipleModels-Vallina | 1.02 ± 0.09 | 0.97 ± 0.10 | 0.93 ± 0.11 | 0.16 ± 0.09 | 0.00 ± 0.00 | **0.62 ± 0.44** |
| MultipleModels-COM | 0.99 ± 0.02 | 0.99 ± 0.08 | 0.43 ± 0.02 | **0.21 ± 0.02** | 0.05 ± 0.09 | 0.53 ± 0.39 |
| MultipleModels-IOM | **1.02 ± 0.02** | **1.01 ± 0.04** | 0.53 ± 0.04 | 0.16 ± 0.08 | 0.10 ± 0.12 | 0.56 ± 0.40 |
| MultipleModels-RoMA | **1.02 ± 0.02** | **1.01 ± 0.04** | 0.53 ± 0.04 | 0.16 ± 0.08 | 0.10 ± 0.12 | 0.56 ± 0.40 |
| MultipleModels-ICT | 0.99 ± 0.02 | 0.99 ± 0.08 | 0.43 ± 0.02 | **0.21 ± 0.02** | 0.05 ± 0.09 | 0.53 ± 0.39 |
| MultipleModels-TriMentoring | 0.99 ± 0.02 | 0.99 ± 0.08 | 0.43 ± 0.02 | **0.21 ± 0.02** | 0.05 ± 0.09 | 0.53 ± 0.39 |
| ParetoFlow-Vallina | 0.70 ± 0.05 | 0.67 ± 0.01 | 0.73 ± 0.04 | 0.00 ± 0.00 | **0.84 ± 0.01** | 0.59 ± 0.30 |
| Diffusion-Guidance-Crowding | 0.71 ± 0.08 | 0.86 ± 0.05 | 0.92 ± 0.06 | 0.00 ± 0.00 | 0.00 ± 0.00 | 0.50 ± 0.41 |
| Diffusion-Guidance-SubCrowding | 0.39 ± 0.07 | 0.18 ± 0.10 | 0.36 ± 0.08 | 0.00 ± 0.00 | 0.00 ± 0.00 | 0.19 ± 0.17 |

Table 10: $\mathrm{MMD}(\hat{P}_{\mathrm{alg}}, \hat{P}_{\mathrm{off}})$ results of DTLZ tasks (↑). Each algorithm is run for 5 seeds and evaluated on 256 designs. Boldface marks the methods with the largest distance.

| | DTLZ1 | DTLZ2 | DTLZ3 | DTLZ4 | DTLZ5 | DTLZ6 | DTLZ7 | Avg. |
|---|---|---|---|---|---|---|---|---|
| MultiHead-Vallina | **0.80 ± 0.06** | **0.41 ± 0.02** | 0.32 ± 0.23 | **0.37 ± 0.02** | **0.59 ± 0.01** | **0.74 ± 0.02** | **0.84 ± 0.10** | **0.58 ± 0.20** |
| MultiHead-PcGrad | 0.46 ± 0.05 | 0.33 ± 0.06 | 0.02 ± 0.04 | 0.06 ± 0.08 | 0.56 ± 0.01 | 0.66 ± 0.02 | 0.83 ± 0.06 | 0.42 ± 0.28 |
| MultiHead-GradNorm | 0.66 ± 0.16 | 0.15 ± 0.14 | **0.33 ± 0.41** | 0.14 ± 0.18 | 0.42 ± 0.27 | 0.64 ± 0.12 | 0.84 ± 0.15 | 0.45 ± 0.25 |
| MultipleModels-Vallina | **0.80 ± 0.06** | **0.41 ± 0.02** | 0.32 ± 0.23 | **0.37 ± 0.02** | **0.59 ± 0.01** | **0.74 ± 0.02** | **0.84 ± 0.10** | **0.58 ± 0.20** |
| MultipleModels-COM | 0.45 ± 0.07 | 0.37 ± 0.01 | 0.00 ± 0.00 | 0.22 ± 0.16 | 0.54 ± 0.02 | 0.59 ± 0.02 | 0.68 ± 0.01 | 0.41 ± 0.22 |
| MultipleModels-IOM | 0.45 ± 0.04 | 0.37 ± 0.02 | 0.00 ± 0.00 | 0.15 ± 0.12 | 0.55 ± 0.02 | 0.57 ± 0.03 | 0.72 ± 0.02 | 0.40 ± 0.23 |
| MultipleModels-RoMA | 0.45 ± 0.04 | 0.37 ± 0.02 | 0.00 ± 0.00 | 0.15 ± 0.12 | 0.55 ± 0.02 | 0.57 ± 0.03 | 0.72 ± 0.02 | 0.40 ± 0.23 |
| MultipleModels-ICT | 0.45 ± 0.07 | 0.37 ± 0.01 | 0.00 ± 0.00 | 0.22 ± 0.16 | 0.54 ± 0.02 | 0.59 ± 0.02 | 0.68 ± 0.01 | 0.41 ± 0.22 |
| MultipleModels-TriMentoring | 0.45 ± 0.07 | 0.37 ± 0.01 | 0.00 ± 0.00 | 0.22 ± 0.16 | 0.54 ± 0.02 | 0.59 ± 0.02 | 0.68 ± 0.01 | 0.41 ± 0.22 |
| ParetoFlow-Vallina | 0.48 ± 0.04 | 0.02 ± 0.05 | 0.00 ± 0.00 | 0.00 ± 0.00 | 0.23 ± 0.13 | 0.51 ± 0.08 | 0.33 ± 0.09 | 0.22 ± 0.21 |
| Diffusion-Guidance-Crowding | 0.54 ± 0.02 | 0.00 ± 0.00 | 0.00 ± 0.00 | 0.04 ± 0.08 | 0.00 ± 0.00 | 0.65 ± 0.04 | 0.73 ± 0.07 | 0.28 ± 0.32 |
| Diffusion-Guidance-SubCrowding | 0.54 ± 0.02 | 0.00 ± 0.00 | 0.00 ± 0.00 | 0.00 ± 0.00 | 0.00 ± 0.00 | 0.33 ± 0.03 | 0.30 ± 0.05 | 0.17 ± 0.21 |

Table 11: MOO metrics on ZDT and DTLZ tasks. Results are reported for the best-performing evolutionary (Evo.) and generative (Gen.) methods. Each algorithm is run for 5 seeds and evaluated on 256 designs. Boldface indicates the best value.

| | | ZDT1 | ZDT2 | ZDT3 | ZDT4 | ZDT6 | DTLZ1 | DTLZ2 | DTLZ3 | DZLZ4 | DTLZ5 | DZLZ6 | DTLZ7 |
|---|---|---|---|---|---|---|---|---|---|---|---|---|---|
| HV (↑) | Evo. | **4.83 ± 0.00** | 5.57 ± 0.05 | 5.59 ± 0.06 | 4.74 ± 0.31 | 4.78 ± 0.00 | 10.64 ± 0.00 | 12.44 ± 0.00 | 9.89 ± 0.00 | 17.70 ± 0.01 | 10.76 ± 0.00 | 10.95 ± 0.01 | 10.74 ± 0.01 |
| | Gen. | 4.53 ± 0.03 | **5.67 ± 0.19** | **5.60 ± 0.05** | **5.03 ± 0.06** | **4.82 ± 0.01** | **10.64 ± 0.00** | 12.39 ± 0.01 | **9.89 ± 0.00** | 17.60 ± 0.03 | 10.59 ± 0.02 | 10.80 ± 0.03 | 9.78 ± 0.08 |
| GD+ (↓) | Evo. | **0.01 ± 0.00** | 0.05 ± 0.03 | 0.17 ± 0.02 | 0.68 ± 0.08 | 0.08 ± 0.07 | 0.25 ± 0.04 | 0.02 ± 0.01 | 0.38 ± 0.16 | 0.18 ± 0.04 | 0.00 ± 0.00 | 0.46 ± 0.01 | 0.08 ± 0.07 |
| | Gen. | 0.29 ± 0.03 | 0.18 ± 0.02 | 0.24 ± 0.02 | **0.39 ± 0.06** | 0.17 ± 0.00 | 0.40 ± 0.01 | 0.16 ± 0.03 | 0.52 ± 0.00 | 0.22 ± 0.00 | 0.16 ± 0.01 | 0.60 ± 0.02 | 0.35 ± 0.02 |
| IGD+ (↓) | Evo. | **0.01 ± 0.00** | **0.04 ± 0.01** | 0.12 ± 0.01 | 0.38 ± 0.14 | 0.02 ± 0.00 | **0.08 ± 0.01** | 0.01 ± 0.00 | 0.09 ± 0.02 | 0.06 ± 0.01 | 0.00 ± 0.00 | 0.40 ± 0.02 | 0.02 ± 0.00 |
| | Gen. | 0.13 ± 0.02 | 0.10 ± 0.02 | 0.10 ± 0.01 | 0.25 ± 0.04 | 0.01 ± 0.01 | 0.09 ± 0.03 | 0.12 ± 0.02 | 0.18 ± 0.02 | 0.20 ± 0.04 | 0.08 ± 0.01 | 0.24 ± 0.05 | 0.19 ± 0.02 |
| MMD (↑) | Evo. | 1.02 ± 0.02 | 1.01 ± 0.04 | 0.95 ± 0.16 | 0.21 ± 0.02 | 0.14 ± 0.26 | 0.80 ± 0.06 | 0.41 ± 0.02 | 0.33 ± 0.41 | 0.37 ± 0.02 | 0.59 ± 0.01 | 0.74 ± 0.02 | 0.84 ± 0.10 |
| | Gen. | 0.71 ± 0.08 | 0.86 ± 0.05 | 0.92 ± 0.06 | 0.00 ± 0.00 | **0.84 ± 0.01** | 0.54 ± 0.02 | 0.02 ± 0.05 | 0.00 ± 0.00 | 0.04 ± 0.08 | 0.23 ± 0.13 | 0.65 ± 0.04 | 0.73 ± 0.07 |

