# OpenReview forum: "The Offline-Frontier Shift: Diagnosing Distributional Limits in Generative Multi-Objective Optimization"
_ICLR.cc/2026/Workshop/Sci4DL — Sci4DL 2026_

### Official Review · Reviewer_Eccj · 2026-02-20

**Fit:** 2
**Significance:** 2
**Confidence:** 2

**Summary:**

This paper analyzes the capability of generative methods to solve the multi-objective optimization problem, which requires sampling solutions that are out-of-distribution in the objective space. They compare with evolutionary methods, and the authors show that while the hypervolume metric suggests competitive performance, this does not show the complete picture, as there are other metrics where evolutionary methods are clearly better. They explain this in terms of an offline-frontier shift, showing that generative models remain close to the training distribution rather than exploring toward the Pareto front.

This is an important problem, and the paper does good work shedding light on it.

**Strengths:**

- The writing is clear.
- The introduction and experiment in terms of offline-frontier shift are well presented.

**Suggestions:**

- The paper would benefit from a brief description of how each class of methods (evolutionary and generative) operates in this offline MOO context, to help readers understand the fundamental difference between the two approaches and why one might be expected to explore more aggressively than the other.
- The paper identifies the offline-frontier shift as a fundamental limitation but does not propose any potential directions. Examples of them might be guided sampling, using reweighting schemes (or even SMC).
A discussion of these points can help to understand where the work is going and the feasibility of extending the workshop paper to a long version.
- The offline-frontier shift s(P_off) requires computing orthogonal projections onto the Pareto front manifold M, but M is generally unknown and must be approximated. The paper does not explain how this projection is computed in practice: whether M is estimated from the data, from a reference set, or assumed to be known. This is a non-trivial step and should be clarified, as the quality of the shift estimate directly affects the validity of the empirical results.

---

### Meta-Review · Area_Chair_2VoE · 2026-03-01

**Recommendation:** Accept

**Metareview:**

This work presents interesting insights into generative optimization and it is a good fit for the workshop.

---

### Decision · Program_Chairs · 2026-03-02

Accept